# Early and adult life environmental effects on reproductive performance in preindustrial women

**Lidia Colejo-Durán** [1,2] *, **Fanie Pelletier**[1], **Lisa Dillon**[3], **Alain Gagnon**[3], **Patrick Bergeron**[2]

**1** Département de Biologie, Université de Sherbrooke, Sherbrooke, Québec, Canada, **2** Department of Biology and Biochemistry, Bishop's University, Sherbrooke, Québec, Canada, **3** Department of Demography, Université de Montréal, Montréal, Québec, Canada

* Lidia.colejo.duran@usherbrooke.ca

**Data Availability Statement:** The data used in this study contains anonymized information about individuals who lived in Nouvelle France. Access to this data is managed by the Programme de

## Abstract

Early life environments can have long-lasting effects on adult reproductive performance, but disentangling the influence of early and adult life environments on fitness is challenging, especially for long-lived species. Using a detailed dataset spanning over two centuries, we studied how both early and adult life environments impacted reproductive performance in preindustrial women. Due to a wide geographic range, agricultural production was lower in northern compared to southern parishes, and health conditions were worse in urban than rural parishes. We tested whether reproductive traits and offspring survival varied between early and adult life environments by comparing women who moved between different environments during their lifetime with those who moved parishes but remained in the same environment. Our findings reveal that urban-born women had an earlier age at first reproduction and less offspring surviving to adulthood than rural-born women. Moreover, switching from urban to rural led to increased offspring survival, while switching from rural to urban had the opposite effect. Finally, women who switched from rural to urban and from South to North had their first child at an older age compared to those who stayed in the same environment type. Our study underscores the complex and interactive effects of early and adult life environments on reproductive traits, highlighting the need to consider both when studying environmental effects on reproductive outcomes.

## Introduction

Environmental effects on phenotypes are particularly pronounced when they occur early in life [1–3], especially because they can cause delayed carry-over effects on traits expressed later in life [4]. The influence of the early life environment on individuals' fitness is multifaceted, including inter-individual variation in the expression of traits that can impact survival and reproduction [2, 5, 6]. In red deer (*Cervus elaphus*), for example, warm springs improved foraging conditions such that cohorts reproducing earlier had higher survival rates as adults than those born during cold springs [7]. Similarly, semi-captive female elephants (*Elephas*

recherche en démographie historique (PRDH) from the Université de Montréal. Researchers can request full dataset access through an approved access request by the PRDH. - Website: https://www.prdh-igd.com/en/home - Public Access: https://www.prdh-igd.com/Gratuit/en/PRDH/Recherche/Acte - Subscriber and institutional access:https://www.prdh-igd.com/Login?ReturnUrl=%2fMembership%2fen%2fPRDH%2fRecherche%2fActe - Contact: contact@institutdrouin.com Additionally, the two subsets used for the analysis shown in the manuscript have been uploaded to Dryad and can be found at the following link, to facilitate the review: https://datadryad.org/stash/share/u3ipgAeBOz-DuLJAIirmE3_733AKXoExPIWICVNgL3w This link can be made public upon publication.

**Funding:** This study was funded by the Natural Sciences and Engineering Research Council of Canada (NSERC) Discovery Grant #2016-04193 to P. B. and #2018-05405 to F. P. (see https://www.nserc-crsng.gc.ca/professors-professeurs/grants-subs/dgigp-psigp_eng.asp). It was also founded by the NSERC E.W.R. Steacie Memorial Fellowship #549146-2020 to F. P. (see https://www.nserc-crsng.gc.ca/prizes-prix/steacie-steacie/index-index_eng.asp). The funders had no role in study design, data collection and analysis, decision to publish, or preparation of the manuscript.

**Competing interests:** The authors have declared that no competing interests exist.

*maximus*) born during stressful months, characterised by heavy workload, experienced faster reproductive senescence and lower reproductive success than those born during non-stressful months [8]. However, it remains a challenge to address these questions on long-lived animals because it requires long-term monitoring of their complete life-histories in the wild.

Applying an evolutionary ecology approach to long-term human datasets has been shown to provide valuable insights for addressing questions in life history evolution [9–11]. Datasets on rural or preindustrial populations are particularly valuable for examining the effects of early and adult life environments on fitness [6]. For instance, rural Gambians born during the yearly "hungry season" faced a risk of premature death in early adult life that was up to ten times higher than those born during other seasons [12]. Parish registers from pre-industrial populations also provide unique information on the geographical locations of major life-history events and individual details such as family size, genealogy, and lifespan. Such data is rarely available for wild animals over extended periods. For example, a study on population range expansion in Québec found that effective fertility was about 20% higher for females living in newly established parishes compared to those in the core [13].

Several hypotheses have been proposed to explain the relative effects of early and adult life environments on fitness related traits, such as the silver spoon [14–16] (also known as "developmental constraints" [17]), the Predictive Adaptive Response (PAR) [18], and the importance of the quality of the adult life environment [14]. Expected fitness effects of these hypotheses are listed in Table 1. Briefly, 1) the silver spoon hypothesis suggests that individuals born in favourable environments should perform better later in life than those born in poorer environmental conditions [19]. Yet, recent studies suggested that silver spoon effects can be non-linear, being stronger when the quality of the adult life environment is intermediate, while most individuals should perform poorly in harsh environmental conditions or do very well in excellent conditions [20, 21]. 2) The PAR hypothesis predicts that a match between the early and adult life environment would maximise fitness, regardless of the environmental quality [18, 22, 23]. Yet, empirical support to the PAR hypothesis remains scarce in long-lived wild species [24]. 3) The fitness of individuals can also be influenced by the quality of the adult life environment, regardless of the early life environment [25–27]. Therefore, it is important to integrate information on the quality of the adult life environment, especially the reproductive environment, when studying the effects of the early life environment on fitness [14, 20, 28–30]. These three hypotheses suggest a link between reproductive performance and environmental conditions, with context-specific long-term impacts of both early and adult life environments [25–27]. These hypotheses are not always mutually exclusive, but rather context-dependent, enabling us to examine which one is operative in the case of pre-industrial humans [14, 21].

Persistent and latent health effects, such as heart disease emerging in middle age, could result from early life biological mechanisms such as epigenomic fetal programming [31]. For example, a detrimental nutrition given by a bad early life environment could affect the age at first reproduction, which is influenced by the age of menarche [32, 33]. Both early and late age at reproduction can have detrimental effect on the mother and her offspring [34–38]. In addition, a bad reproductive environment has been shown to shorten interbirth intervals, by reducing the period of lactational amenorrhea [33]. It is therefore important to study the long-term effects of both early and adult life environments, as they can affect the health of the individual [31], and consequently affect their reproductive outcomes [39].

To assess the influence of early and adult life environments on reproductive performance, studies typically compare individuals born in different seasons or years while living at the same location [7, 8, 40–42]. Although temporal changes in environmental conditions are of great interest [41, 43], they are likely to be less pronounced compared to the habitat changes experienced by individuals that disperse. However, obtaining data on animal dispersion and

**Table 1. Predictions on fitness based on early and late life environmental conditions, as expected by the hypotheses of silver spoon, Predictive Adaptive Response (PAR), and quality of adult life environment.**

| Environment | | Predictions on fitness | | |
|---|---|---|---|---|
| *Early* | *Late* | *Silver Spoon* | *PAR* | *Quality of Adult Life Environment* |
| *Good* | *Good* | + | + | + |
| *Good* | *Bad* | + | – | – |
| *Bad* | *Good* | – | – | + |
| *Bad* | *Bad* | – | + | – |

"+" predicts a positive outcome, while "–" predicts a negative one.

the impact on lifetime reproductive success in wild populations is challenging due to the need for comprehensive longitudinal data on reproduction throughout the entire lifespan [44]. One of the largest databases on preindustrial humans comes from the *Registre de la population du Québec Ancien* (RPQA) [45]. This dataset covers about the first two centuries of settlement in Nouvelle-France (current Province of Québec, Canada), and includes almost half a million individuals with reconstructed life history over nine generations [45], making it an excellent resource for studying environmental effects on fitness.

In this study, we examined how early and adult life environments influenced the lifetime reproductive performance of preindustrial women. Previous studies indicated that rural parishes and the south shore of the Saint Lawrence River provided better environmental conditions—such as greater food availability—compared to urban parishes and the north shore [42, 46, 47] (See S1 Fig). To isolate the impacts of early and adult life conditions, we studied the life history of women who changed parishes between their own birth and the birth of most of their offspring. By including ecological characteristics of the parishes, we attempted to distinguish the effects of early life conditions from those resulting from changes in the adult life environment (urban/rural, north/south shore). We also aimed at investigating whether the environmental effects on reproductive performance were better explained by the silver-spoon hypothesis, the PAR, or the environmental quality during adult life.

## Materials and methods

### Study population and historical context

The RPQA database includes 448,501 individuals and 74,067 families [45], from 1572 (earliest date of birth of a settler, before migrating to the colony) to 1870 (latest date of death of an individual) in 166 different parishes (S1 Fig). It was assembled by reconstituting family histories from Roman Catholic parish registers of baptisms, marriages, and burials from which the date and location of the events are known [45, 48, 49]. We restricted this database to women who were born in Nouvelle-France, with known parents, with known date and location of birth and death (n = 216,442), and who did not remarry (93.6% of women). As we were interested in contrasting early and adult life environments, we also only included women who had relocated to a different parish between their birth and reproduction. In addition, we selected two subsets, one with women born between 1640 and 1750, which contains 7,203 women who had 61,606 offspring, of whom 29,059 survived to adulthood, and another with women born between 1640 and 1729, which contains 3,959 women who had 33,878 offspring, of whom 19,865 survived to adulthood (see S2 Fig for more details on filtering). This filtering ensured that all women expressed a complete reproductive history, since the date of death of the descendants is less precise at the end of the record (see also [50]). For further descriptive statistics of the subsets, see S1–S3 Tables.

During the study period, there was limited emigration from the colony, such that the population is considered semi-closed [45, 48, 49, 51]. The population was composed primarily of farmers, with a smaller number of artisans, merchants, civil servants, professionals, and elites residing in urban areas [52]. Québec City (founded in 1608) and Ville-Marie (founded in 1642, now known as Montréal) were the only urban parishes along the St. Lawrence River [52–54]. Nouvelle-France experienced lower overall mortality rates compared to Europe at the same time [55–57], although infant mortality rates were similarly high [58]. The population was characterised by a high fertility rate compared to other preindustrial human populations [45], which led to exponential growth [59, 60], with the average family size in our subset being 8.6 (S1 Table). Extramarital relationships were socially condemned and very rare; extra-conjugal paternity has been estimated to be less than 1% [61, 62].

## Environmental conditions

The environmental characteristics of the area were distinctive, with the north and south shores of the St. Lawrence River differing in average temperatures, persistence of snow cover, and soil fertility [63]. For example, the frost-free season was about one month longer in the south-west than in the north-east of the study area [42]. According to the analysis of the 1688 Census of New-France (Statistics Canada) conducted by Gagnon (2012), in the southern regions, each inhabitant had approximately 2.5 acres (1.01 hectares) of land under cultivation, whereas in the north, the corresponding figure was 2.2 acres (0.89 hectares) per inhabitant. Not only did inhabitants in the south possess larger plots of land, but they also had larger agricultural yields, particularly in wheat production. It is estimated that the southern lands yielded 5.8 bushels (204 litres) of grain per acre sown, whereas the northern lands yielded 4.9 bushels (173 litres) per acre [42]. Consequently, the south reported a grain yield of 14.5 bushels (511 litres) per inhabitant from the previous fall's harvest, whereas the north yielded 10.7 bushels (377 litres) per inhabitant in 1688 [42]. Thus, inhabitants in the southern part of the St. Lawrence River Valley experienced better agricultural conditions and their farms were more productive than those from the North [42].

In addition, mortality patterns differed between urban and rural populations due to socio-economic factors [52], given that rural inhabitants had greater food availability, lower exposition to diseases and lower infant mortality than urban populations [46, 59]. Also, rural-born women and men were married in slightly greater proportions than urban-born ones [64]. Epidemic cycles were also common in both Québec City and Montréal because of the increasing urban population density and poor hygiene [57, 65]. Women who lived in Québec City and Montréal were thus classified as urban residents and the rest of the population was designated as rural, with urban parishes being considered a worse environment than rural parishes.

The parish for early life environment was characterised following previous papers on the same population [42, 50] and we described it with the variable *Birth Environment*, with four levels: Rural South (both good environments), Rural North (good urbanity, bad shore), Urban South (bad urbanity, good shore) and Urban North (both bad environments). To assess the impact of switching environments between early and adult life, we also include the variables *Switching Urbanity* and *Switching Shore*. *Switching Urbanity* indicated whether women stayed in the same urbanity while *Switching Shore* indicated whether they stayed on the same shore. Both switching variables had three levels: same environment or transition from bad to good or good to bad environments. Consequently, we determined the early-life environment based on the individual's parish of birth and the adult life environment based on the type of environment where the greatest proportion of the individual's offspring were born, as our interest lies in their reproductive environment. For example, if a woman was born in Montréal, married at

19 in the same parish, had her first child at 21 in Québec City, and then had her remaining six children in a rural parish on the north shore of the St. Lawrence River, we would have categorized her early-life environment as Urban South and her adult life environment as Rural North. Therefore, she would have switched urbanity from Urban to Rural and shore from South to North. The variables for switching allowed us to examine the effect of a transition between the early and the adult life environments. We also considered the effect of the distance between the early and adult life parishes, as it has already been shown to influence fitness [50]. We calculated it as the Haversian distance (km) between the geographic coordinates of the parish of birth and the parish of first reproduction, using the *geosphere* package [66].

### Reproductive traits

We used three important traits to quantify women's reproductive performance: age at first reproduction (AFR), defined as the age at which a woman first reproduced, number of offspring (NO), corresponding to the number of infants born to each mother, and lifetime reproductive success (LRS), corresponding to the number of children who survived to at least 15 years of age, considered to be the age of maturity in preindustrial human populations (e.g. [50, 67]). Survival to 15 was established by the presence of subsequent life events when the date of death was unknown [50]. Better reproductive performance would be associated with a younger Age at First Reproduction (AFR), along with higher numbers of offspring (NO) and lifetime reproductive success (LRS).

We considered two additional traits in our analysis: *Fertile years*, also known as reproductive tenure [68, 69], which refers to the period from marriage until the death of either spouse or until both partners reached 45 years old [50]. This trait represents the years during which a woman had the opportunity to reproduce. *Fertile years* was included as a covariate in the analyses of NO and LRS and also as response variable in a model that we report in supplementary material (S4 and S5 Tables, S3 and S4 Figs). Furthermore, we examined the proportion between LRS and NO (LRS/NO). Since the results for LRS/NO were qualitatively similar to those for LRS (see below), they are reported in the supplementary materials (S4 Table, S3 and S4 Figs) and will not be discussed further.

We also conducted a sensitivity analysis on Age at Marriage, which is a strong determinant of the onset of reproduction [70], to validate our results on AFR (S2 Appendix, S4 Table and S5 Fig). From our data, first births typically occurred around 1.2 years after the wedding, and the findings regarding age at marriage for the different environments mirrored those for AFR. Additionally, we performed a sensitivity analysis for all models by excluding the women who had their first reproduction before the age of 15, to ensure that our conclusions were robust against the potential biases associated with very young mothers [34, 38]. Since only 0.21% of women in our filtered dataset gave birth before 15 years old, excluding them (analyses not shown) did not change our conclusions.

### Statistical analyses

We ran three sets of models, one for each reproductive trait. AFR was normally distributed and analysed with a linear mixed model (LMM). NO and LRS were analysed with a generalised linear mixed model (GLMM) with a Poisson distribution. The models accounted for early life environment with the variable *Birth Environment* and for environmental switch between birth and adulthood with the variables *Switching Urbanity*, *Switching Shore*, and the interaction between them. Other fixed effects included in the model were *fertile years*, *wave front* and *period*. *Fertile years* was only included in the models for NO and LRS. *Wave front* has been shown to influence family size [13] and was calculated as the difference between a life-history

event (first reproduction in our case) and the date of the foundation of the parish of birth. The variable *Period* was defined as categorical and included five intervals [71] of 20 years, except the first one which spans 30 years to maintain a balanced sample size. Women were categorized based on their date of birth: 1640–1669, 1670–1689, 1690–1709, 1710–1729 and after 1730. The fifth period does not appear in the analysis for LRS, since this subset only included women born before 1729. All models incorporated the random terms *family identity* (marriage ID of the woman's parents), to control for genetic and shared family environment effects, and *year of birth*, to account for potential cohort effects [71]. Therefore, the *Period* variable accounted for changes that may have occurred in the environment or population across different time intervals, while *year of birth* controlled for yearly events such as poor harvests or epidemics.

All analyses were performed using R version 4.3.1 with the package *glmmTMB* [72]. We presented the full models on Table 2 and S4 Table. Careful consideration was given to the selection of variables included in those models to ensure a robust and interpretable outcome. An additional set of models is presented in S6 Table, sequentially adding the fixed effects related to the environment. The marginal and conditional $R^2$ of these models is presented in S7 Table.

We calculated the conditional and marginal $R^2$ for the models with the package *MuMIn* [73] and for each variable we calculated the partitioned $R^2$ with the package *glmm.hp* [74]. We also performed post-hoc Tukey tests with the package *emmeans* [75] to conduct pairwise multiple comparisons of environmental effects on reproductive traits.

## Results

### Age at first reproduction *(AFR)*

Both the early life environment (*Birth Environment*: $\chi^2$ [3] = 112.44, p <0.001) and the environmental switch between birth and adulthood (*Switching Urbanity * Switching Shore*: $\chi^2$ [4] = 6.65, p = 0.156) were related to AFR, as well as the *wave front* and *distance* (Table 2). Compared to women born in rural areas, those born in urban areas had their first child on average 3.5 years earlier if born on the south shore and 4.9 years earlier if born on the north shore (Fig 1A and S8 Table). These differences were also significant according to the post-hoc Tukey test. In addition, women born in the southern urban parishes began reproducing 1.4 years later than those born in the northern urban ones, which was also significant according to the Tukey test (Fig 1A and S8 Table). For the interaction *Switching Urbanity*Switching Shore*, women who remained in the same type of urbanity reproduced between 1.4 and 1.5 years earlier, depending on the shore switch, compared to those who switched from rural to urban areas and from south to north (Fig 2 and S9 Table). Switching from rural to urban areas and from south to north was the only one significant after the Tukey test. *Distance* between early and adult life environments also influenced AFR (Table 2); women who moved greater distances began to reproduce later (S6A Fig). Overall, 12% of the variance in AFR was explained by the fixed effects of the final model (S5 Table), and most of the variance explained was due to the *wave front*, followed by the early life environment (S5 Table).

### Number of Offspring *(NO)*

The number of offspring born varied positively with the number of *fertile years* of the mother, was negatively related to the number of years since a parish was established *(wave front)* and varied slightly with *Period* ($\chi^2$ [4] = 20.77, p <0.001, Table 2). The variables for early life environment (*Birth Environment*: $\chi^2$ [3] = 12.69, p = 0.386) and environmental switch (*Switching Shore*: $\chi^2$ [2] = 4.48, p = 0.106; *Switching Urbanity*: $\chi^2$ [2] = 0.12, p = 0.942) were not significant

**Table 2. Output of the full models with all the variables present in the first column, for Age at First Reproduction (AFR), Number of offspring (NO), and Lifetime reproductive success (LRS).**

| Variables | | AFR | | | NO | | | LRS | | |
|---|---|---|---|---|---|---|---|---|---|---|
| | | Estimate | SE | P value | Estimate | SE | P value | Estimate | SE | P value |
| (Intercept) | | 23.491 | 0.397 | <*0.001* | 1.878 | 0.032 | <*0.001* | 1.535 | 0.047 | <*0.001* |
| **Birth Environment** | Rural North | -0.050 | 0.140 | *0.723* | -0.012 | 0.011 | *0.308* | -0.004 | 0.020 | *0.861* |
| | Urban South | -3.571 | 0.529 | <*0.001* | 0.050 | 0.044 | *0.255* | -0.320 | 0.075 | <*0.001* |
| | Urban North | -4.935 | 0.491 | <*0.001* | 0.026 | 0.040 | *0.518* | -0.303 | 0.075 | <*0.001* |
| **Fertile years** | | — | — | — | 0.418 | 0.005 | <*0.001* | 0.461 | 0.009 | <*0.001* |
| **Wave front** | | 1.974 | 0.082 | <*0.001* | -0.027 | 0.007 | <*0.001* | -0.042 | 0.014 | **0.002** |
| **Distance** | | 0.166 | 0.059 | **0.005** | 0.003 | 0.005 | *0.554* | 0.003 | 0.008 | *0.742* |
| **Period** | 1670–1689 | 1.359 | 0.378 | <*0.001* | 0.132 | 0.029 | <*0.001* | 0.118 | 0.038 | **0.002** |
| | 1690–1709 | 1.690 | 0.380 | <*0.001* | 0.190 | 0.030 | <*0.001* | 0.109 | 0.041 | **0.008** |
| | 1710–1729 | 0.895 | 0.388 | **0.021** | 0.205 | 0.030 | <*0.001* | -0.048 | 0.045 | *0.286* |
| | 1730–1750 | -0.563 | 0.400 | *0.159* | 0.193 | 0.031 | <*0.001* | — | — | — |
| **Switching Urbanity** | Urban to Rural | 0.276 | 0.484 | *0.568* | -0.003 | 0.040 | *0.937* | 0.288 | 0.069 | <*0.001* |
| | Rural to Urban | 0.404 | 0.183 | **0.028** | -0.005 | 0.016 | *0.733* | -0.332 | 0.028 | <*0.001* |
| **Switching Shore** | North to South | 0.033 | 0.170 | *0.848* | 0.028 | 0.014 | **0.043** | 0.007 | 0.023 | *0.777* |
| | South to North | -0.109 | 0.192 | *0.569* | -0.008 | 0.016 | *0.613* | -0.067 | 0.030 | **0.025** |
| **Switching Urbanity * Switching Shore** | Urban to Rural, North to South | 0.354 | 0.426 | *0.406* | -0.005 | 0.035 | *0.893* | 0.064 | 0.055 | *0.244* |
| | Rural to Urban, North to South | 0.058 | 0.448 | *0.897* | -0.039 | 0.038 | *0.301* | 0.029 | 0.063 | *0.647* |
| | Urban to Rural, South to North | 0.446 | 0.537 | *0.406* | 0.009 | 0.044 | *0.841* | -0.005 | 0.075 | *0.948* |
| | Rural to Urban, South to North | 1.119 | 0.468 | **0.017** | -0.068 | 0.041 | *0.097* | -0.138 | 0.074 | *0.064* |

The reference level is "Rural South" for "Birth Environment", "Same Urbanity" for "Switching Urbanity", "Same Shore" for "Switching Shore" and '1640–1669' for "Period". The variable "fertile years" was not included in the AFR model as fixed effect. "—" means that the variable was not included in the full model. P values in bold are significative (p <0.05).

(S3 and S4 Figs). The *distance* variable was not significant (Table 2). Although the fixed effects in this model explained a moderate amount of variation (marginal $R^2$ = 0.57), most of the variation was attributed to the number of fertile years (S5 Table).

## Lifetime Reproductive Success *(LRS)*

Lifetime reproductive success was related to the early life environment (*Birth Environment*: $\chi^2$ [3] = 20.15, p <0.001). Women born in the south urban parish had 1.37 less offspring surviving to adulthood, and those born in the north urban parish had 1.35 less offspring who survived adulthood, than those born in rural parishes (Fig 1B and S7 Table). There was no significant difference in LRS between women born in the south shore compared to the north shore for the same urbanity type (Fig 1B and S8 Table). Despite *Switching Urbanity* *Switching Shore* not being significant ($\chi^2$ [4] = 5.42, p = 0.246), the switch in urbanity (*Switching Urbanity*: $\chi^2$ [2] = 231.62, p <0.001) and in shore (*Switching Shore*: $\chi^2$ [2] = 11.63, p = 0.003) were significant in the model without the interaction. Women who switched from rural to urban (good to bad) had 1.4 less offspring surviving to adulthood than women who stayed under the same conditions and 1.9 less than women who switched from urban to rural (bad to good), (Fig 3A and S10 Table). Women who switched from south to north (good to bad) had on average 1.1 less offspring surviving to adulthood compared to the ones who stayed in the same shore or who moved from north to south (bad to good), (Fig 3B and S11 Table). *Fertile years*, *wave front* and *period* were significant predictors of LRS (Table 2). The marginal $R^2$ of the final

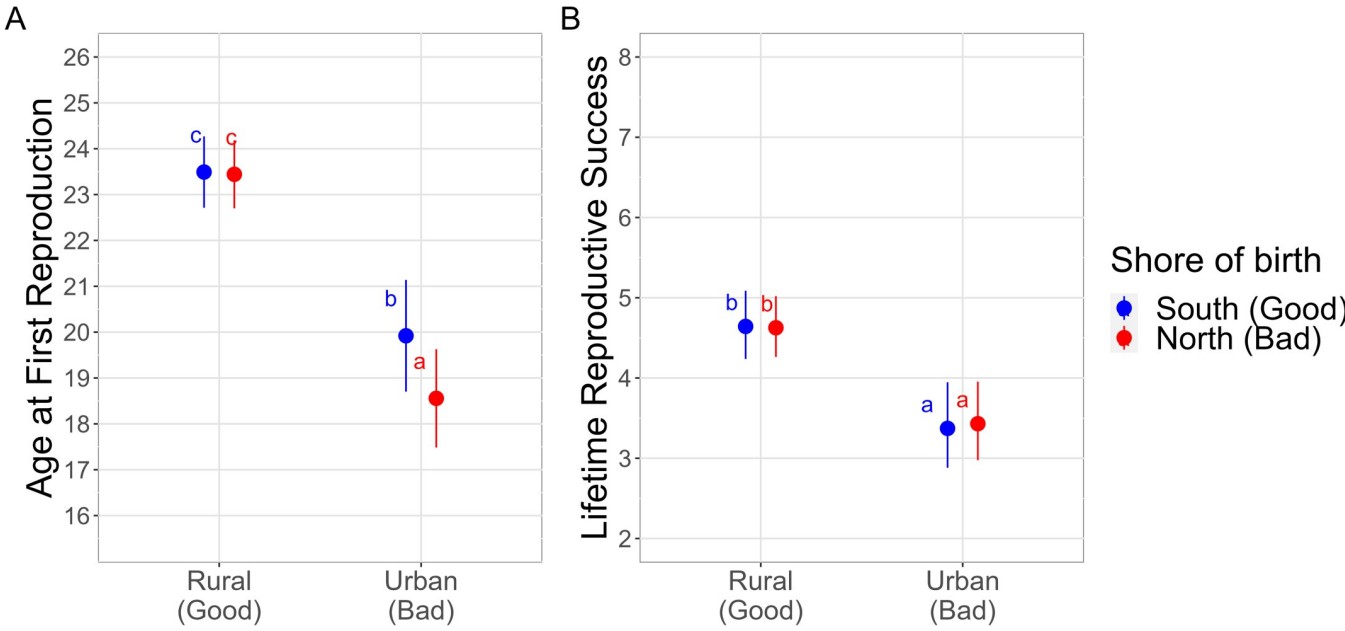

**Fig 1.** Early life environmental effects on Age at First Reproduction (panel A) and Lifetime Reproductive Success (panel B) according to the environment of birth, given by the urbanity (Rural or Urban) and the shore (North and South). Rural and South are considered good environments and Urban and North are considered bad environments. The dots are the predicted marginal values, and the lines are their confidence intervals. Estimates with different letters are statistically different, given by a post-hoc Tukey's pairwise multiple comparison test (P <0.05). See S8 Table for the estimates.

model was 0.55 and most of the variation was explained by the number of *fertile years*, followed by whether a woman switched urbanity and the *period* (S5 Table).

## Discussion

By capitalising on a preindustrial human population dataset characterised by high natural fertility prior to demographic transition [76–78], we disentangled the relative contribution of early and adult life environments to variations in reproductive traits. Given that all the women in our dataset relocated parishes between their birth and reproductive events, we have, at least partially, controlled for the cost of migration. Our study showed that both the early and adult life environments affected reproductive traits, but theoretical explanations of these effects varied by trait. For LRS, our findings support the silver spoon hypothesis, while the results for AFR and LRS also align with the Predictive Adaptive Response (PAR) hypothesis. Additionally, the quality of the adult life environment influenced LRS. NO was not directly affected by the early or the adult life environment.

The silver spoon hypothesis states that individuals born in a good environment will systematically have a better performance than those born in a bad environment ([14–16], (see Table 1: good/good and good/bad). Our study found that women born in rural parishes (good environment) had a higher LRS, consistent with findings from other preindustrial populations, such as the Finnish, for whom better early-life nutrition was associated with improved reproductive performance [79]. Increased malnutrition and a higher prevalence of infections during early life have been shown to result in higher neonatal mortality rates among the offspring of stunted mothers [80], a pattern also observed in some wild animal populations [81]. Different early life nutrition and environmental factors distinctly can impact oocyte and embryo development, prenatal survival, and offspring number and quality [82]. This variation may explain why different environmental characteristics, such as the shore or urbanity of the parish of

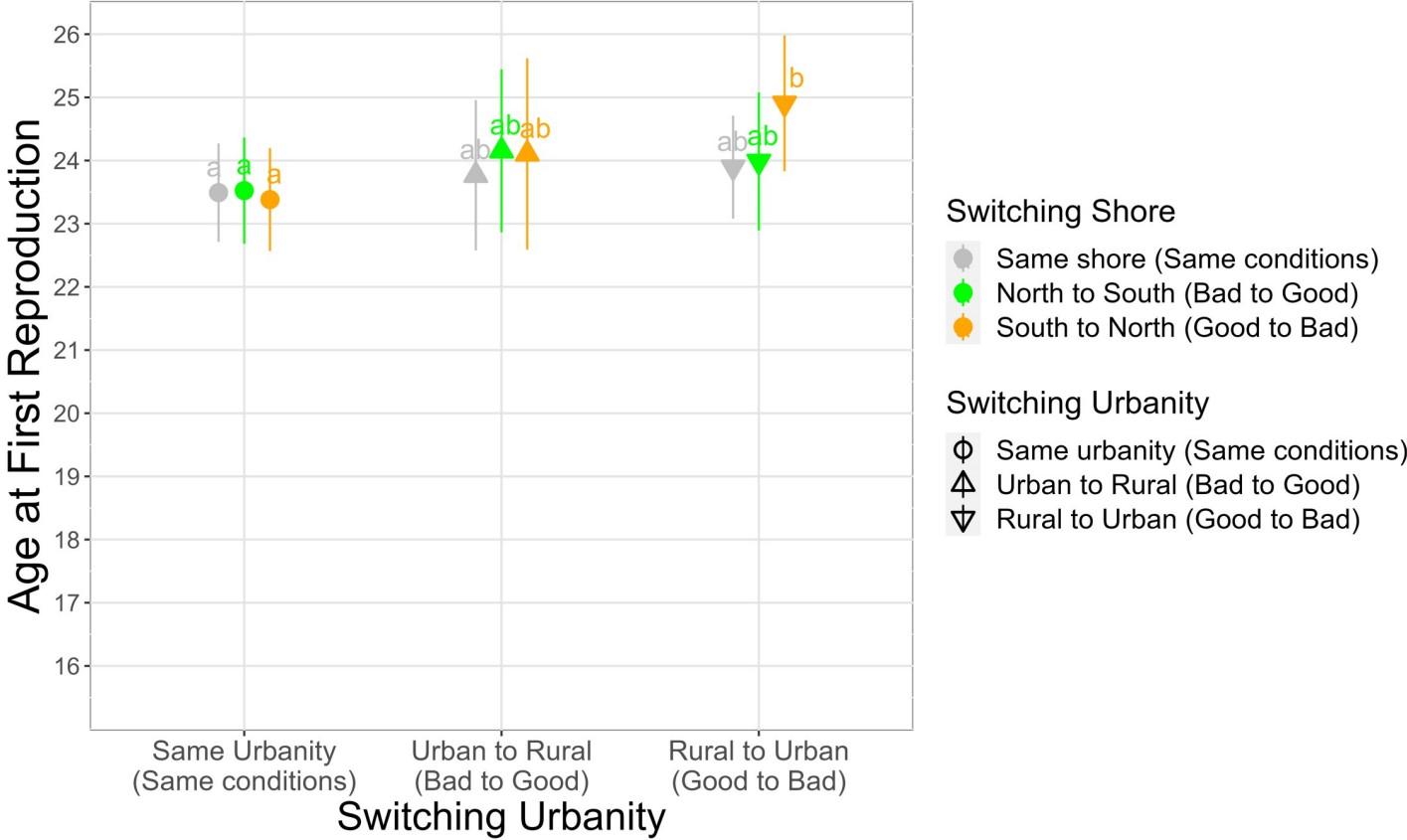

**Fig 2. Adult life environmental effects on Age at First Reproduction according to the interaction between the switch in urbanity and the switch in shore.** Staying in the same urbanity or in the same shore is considered as staying under the same conditions, while switching from urban to rural or north to south is seen as going from a bad to a good environment, and switching from rural to urban or south to north is considered going from a good to a bad environment. The dots are the predicted marginal values, and the lines are their confidence intervals. Estimates with different letters are statistically different, given by a post-hoc Tukey's pairwise multiple comparison test (P <0.05). See S9 Table for the estimates.

birth, distinctly affected reproductive traits like NO or LRS. To better understand the indirect impacts of the environment on NO and LRS, we ran additional models setting the variable representing the number of *fertile years* as the response variable. Despite starting reproduction earlier, women born in urban parishes do not have longer reproductive lifetimes than their rural counterparts, which might explain why they do not have more offspring. This is opposite to findings on long-lived mammal females, for which a longer reproductive lifespan and earlier age at maturity improved lifetime reproductive success [71, 83].

We also found evidence supporting the PAR hypothesis by the joint influence of early and adulthood environments on AFR and LRS. According to the PAR hypothesis, individuals whose adult life environment matches their early life environment are expected to perform better [18, 22, 23] (see Table 1: good/good and bad/bad). For AFR, women who remained in the same urbanity started reproducing earlier than those switching environments. Mechanisms of anticipatory environment matching may have helped ancestral survival amid environmental changes, favouring early reproductive success [18]. Additionally, staying in the same urbanity led to a higher LRS than switching from rural to urban, providing further support for the PAR hypothesis. Previous studies on the Dutch famine found no evidence of PAR on annual reproductive success, but different environments were given by the year of birth instead of by the early life environmental context, like parish type [40]. While we did not find

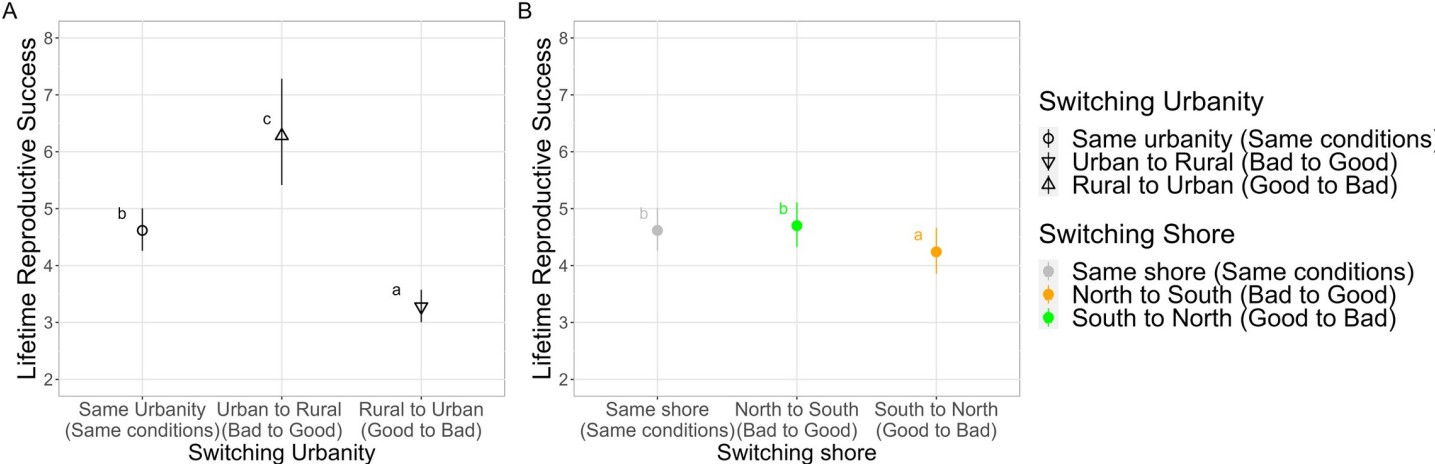

**Fig 3.** Adult life environmental effects on Lifetime Reproductive Success, according to the switch in urbanity (Panel A) and the switch in shore (Panel B). Staying in the same urbanity or in the same shore is considered as staying under the same conditions, while switching from urban to rural or north to south is seen as going from a bad to a good environment, and switching from rural to urban or south to north is considered going from a good to a bad environment. The dots are the predicted marginal values, and the lines are their confidence intervals. Estimates with different letters are statistically different, given by a post-hoc Tukey's pairwise multiple comparison test (P <0.05). See S10 and S11 Tables for the estimates.

support of PAR on NO, we observed that women who remained in the same urbanity had significantly more *fertile years* than those who switched from rural to urban environments (Table 1). Therefore, our results for *fertile years* provide further support for the PAR hypothesis and offer additional insights into the complexity that indirect early life environmental effects can have.

The hypothesis of the importance of the quality of the adult life environment posits that individuals living in a favourable environment during adulthood should display better performance irrespectively of their early life environments (see Table 1: bad/good, good/good) [14, 20, 28–30]. We observed that only the women who moved to urban north (worst environment in our study) experienced detrimental effect on their AFR [20]. Regarding urbanity changes, higher LRS was achieved by women who moved from urban to rural environments. This outcome underscores the importance of the adult life environment, as rural parishes typically have better sanitary conditions [57, 66]. This finding aligns with epidemiological evidence linking poor maternal nutrition during conception or early pregnancy to increased rates of premature birth and offspring health issues [28, 84, 85].

In summary, both the PAR hypothesis and the quality of the adult life environment are important for LRS and AFR. However, for LRS, the early life environment has a greater impact than switching shores, but less impact than moving to urban areas. Therefore, the adult life environment has the strongest influence on LRS, followed by the silver spoon hypothesis and the PAR hypothesis. For AFR, while both the PAR hypothesis and the adult life environment are important, it is difficult to statistically differentiate between them. Nonetheless, the early-life environment has a larger effect on AFR than environmental changes in adulthood. In addition, the random effects contributed further to the variance in AFR than in NO and LRS, indicating that family background had a substantial impact on AFR but much less on NO and LRS.

While our study provides valuable insights into various facets of preindustrial human societies, there were inherent limitations to our approach. First, we were unable to track all the locations where the women lived between the recording of major life history events in the registers. Second, significant familial variation existed, particularly in terms of wealth, for which socioeconomic status information was unavailable [45]. Although we attempted to

account for some of this variability by including family identity as a random effect in the models, fully capturing the diverse circumstances of individuals remains challenging. Third, distinguishing the effects attributed to PAR from those linked to the quality of the adulthood environment was complicated. For example, the detrimental effects on AFR and fertile years were only observed when the women switch to the urban north parish, which is the worst combination. Therefore, we cannot determine if the detrimental effect is caused by switching to a poor adult life environment, which would support the hypothesis of the quality of the adulthood environment, or if it is caused the environmental switch itself, which would support the PAR hypothesis. Fourth, separating the influence of the adulthood environment on the mothers from the early life environment on the offspring was difficult. This distinction is particularly important given the higher child mortality observed in urban parishes within our population [58], which aligns with the lower LRS observed in mothers who switched from rural to urban environments. Pre-industrial human datasets offer unique opportunities to address demographic and eco-evolutionary questions, and we encourage researchers to continue interpreting those historical data with careful consideration of their complexities.

## Conclusion

This study investigates the influence of environments at different life stages on the reproductive performance of preindustrial women. While considering variations in environmental conditions across different periods, we used information on the urbanity and shore of the early life environment, as well as the environmental switch between early and adult life, to characterize the conditions experienced by women with complete reproductive histories. This approach allowed us to separate the environmental effects attributed to the early life environment from those caused by the adult life environment, contributing to a more comprehensive understanding of the relationship between environmental factors and reproductive outcomes. Our findings highlight the importance of both early and adult life environments on human reproductive performance, mainly supporting the silver spoon hypothesis, but also the PAR hypothesis and the importance of the quality of the adult life environment. We hope this approach will inspire colleagues to integrate demography, public health, and evolutionary biology to address interdisciplinary questions.

## Supporting information

**S1 Appendix. Additional analyses.**
(DOCX)

**S2 Appendix. Sensitivity analysis.**
(DOCX)

**S3 Appendix. References of the supporting information.**
(DOCX)

**S1 Fig. Geographical location of the 166 parishes of Nouvelle-France, established between 1616 and 1799.** Panel A shows the location of the parishes along the St. Lawrence Valley, panel B shows a zoom into the area near Québec City and panel C shows a zoom into the area near Montréal. Conditions in the rural parishes were different from those in Québec City and Montréal, and conditions on the north shore (points in red) were also different from those in the south (points in blue). Conditions in the two cities were also different from each other. Modified from "Base de données géographiques et administratives, Données Québec", under a CC BY 4.0 license, with permission from Données Québec, original copyright 2019.
(TIF)

**S2 Fig. Summary of the filters applied on the original dataset (N = 448,501) to obtain the subset used to perform the analyses shown in the manuscript (N = 7,203).** It also indicates which subset was used to calculate the number of offspring (NO) and lifetime reproductive success (LRS).
(TIF)

**S3 Fig.** Early life environmental effects on the Number of Offspring (Panel A), Fertile Years (Panel B) and the Proportion between LRS and NO (panel C) according to the environment of birth, given by the urbanity (Rural or Urban) and the shore (North and South). Rural and South are considered good environments and Urban and North are considered bad environments. The dots are the predicted marginal values, and the lines are their confidence intervals. Estimates with different letters are statistically different, given by a post-hoc Tukey's pairwise multiple comparison test (P <0.05).
(TIF)

**S4 Fig.** Adult life environmental effects on the Number of Offspring (Panels A and B), Fertile years (Panels C and D) and the Proportion between LRS and NO (Panel E) according to the switch in urbanity, the switch in shore, and the interaction between them. Staying in the same urbanity or in the same shore is considered as staying under the same conditions, while switching from urban to rural or north to south is seen as going from a bad to a good environment, and switching from rural to urban or south to north is considered going from a good to a bad environment. The dots are the predicted marginal values, and the lines are their confidence intervals. Estimates with different letters are statistically different, given by a post-hoc Tukey's pairwise multiple comparison test (P <0.05).
(TIF)

**S5 Fig.** Early life environmental effects (Panel A) according to the environment of birth, given by the urbanity (Rural or Urban) and the shore (North and South) and adult life environmental effects (Panel B), according to the interaction between the switch in urbanity and the switch in shore, on Age at Marriage. Rural and South are considered good environments and Urban and North are considered bad environments. Staying in the same urbanity or in the same shore is considered as staying under the same conditions, while switching from urban to rural or north to south is seen as going from a bad to a good environment, and switching from rural to urban or south to north is considered going from a good to a bad environment. The dots are the predicted marginal values, and the lines are their confidence intervals. Estimates with different letters are statistically different, given by a post-hoc Tukey's pairwise multiple comparison test (P <0.05).
(TIF)

**S6 Fig.** Effects of the distance between the parish of birth and the parish of first reproduction on reproductive performance, given by Age at First Reproduction (Panel A), the Proportion between LRS and NO (Panel B) the Age at Marriage (Panel C). The blue lines are the predicted marginal values, and the shaded areas describe the confidence interval.
(TIF)

**S7 Fig. Correlation between reproductive performance and reproductive scheduling, for N = 7,203.** Panel A shows the relationship between the age at first reproduction (AFR) in years and the number of offspring (NO) in terms of children born. Panel B displays the correlation between AFR and lifetime reproductive success (LRS) in terms of the number of children who survived to adult life. Panel C illustrates the correlation between the number of offspring (NO)

and the fertile years in years. Finally, Panel D exhibits the correlation between the fertile years and lifetime reproductive success in terms of the number of children who survived to adult life.
(TIF)

**S1 Table. Descriptive statistics of the means of all the reproductive traits analysed.** They are calculated for N = 7,203, except Lifetime Reproductive Success which is calculated for N = 3,959.
(DOCX)

**S2 Table. Descriptive statistics on the distribution of the population according to the environment of birth, for N = 7,203.**
(DOCX)

**S3 Table. Descriptive statistics on the distribution of the population according to the environmental switch, for N = 7,203.**
(DOCX)

**S4 Table. Output of the full models with all the variables present in the first column, for fertile years, the proportion between Lifetime Reproductive Success and Number of Offspring (LRS/NO) and the age at marriage.**
(DOCX)

**S5 Table. Contributions of each fixed effect for each full model, determined by using the "glmm.hp" package.** These contributions were calculated by hierarchically partitioning the marginal $R^2$ of the generalized mixed-effect model, based on the output from the r.squaredGLMM() function in the "MuMIn" package.
(DOCX)

**S6 Table. Sequential models.**
(DOCX)

**S7 Table. Marginal and conditional R2 of the sequential models, for Age at First Reproduction (AFR), Number of offspring (NO), and Lifetime reproductive success (LRS).**
(DOCX)

**S8 Table. Pairwise comparison between the different categories of Birth Environment for Age at First Reproduction (AFR) and Lifetime reproductive success (LRS).**
(DOCX)

**S9 Table. Pairwise comparison between the different categories of the interaction between Switching Urbanity and Switching Shore for Age at First Reproduction (AFR).**
(DOCX)

**S10 Table. Pairwise comparison between the different categories of Switching Urbanity for Lifetime reproductive success (LRS).**
(DOCX)

**S11 Table. Pairwise comparison between the different categories of Switching Shore for Lifetime reproductive success (LRS).**
(DOCX)

**S1 Data. Models.**
(HTML)

**S2 Data. Partitioned R.**
(HTML)

**S3 Data. Plots.**
(HTML)

**S4 Data. Sequential models.**
(HTML)

## Acknowledgments

We extend our sincere appreciation to the demographers, genealogists, and computer programmers, both past and present, who have dedicated their efforts to RPQA. In addition, we would like to thank Guillaume Blanchet and Dany Garant for their support and expertise, which have improved the manuscript.

## Author Contributions

**Conceptualization:** Lidia Colejo-Durán, Fanie Pelletier, Patrick Bergeron.

**Data curation:** Lidia Colejo-Durán, Lisa Dillon, Alain Gagnon.

**Formal analysis:** Lidia Colejo-Durán.

**Funding acquisition:** Fanie Pelletier, Patrick Bergeron.

**Methodology:** Lidia Colejo-Durán, Fanie Pelletier, Patrick Bergeron.

**Software:** Lidia Colejo-Durán.

**Supervision:** Fanie Pelletier, Patrick Bergeron.

**Validation:** Lidia Colejo-Durán.

**Visualization:** Lidia Colejo-Durán.

**Writing – original draft:** Lidia Colejo-Durán.

**Writing – review & editing:** Fanie Pelletier, Lisa Dillon, Alain Gagnon, Patrick Bergeron.

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
