## [Decision Letter · Decision Letter 0]

1 Feb 2024

PONE-D-23-24394Environmental effects on reproductive performance in preindustrial womenPLOS ONE

Dear Dr.Lidia,

Thank you for submitting your manuscript to PLOS ONE. After careful consideration, we feel that it has merit but does not fully meet PLOS ONE’s publication criteria as it currently stands. Therefore, we invite you to submit a revised version of the manuscript that addresses the points raised during the review process.

We look forward to receiving your revised manuscript.

Kind regards,

Edison Arwanire Mworozi, M.D

Academic Editor

PLOS ONE

Journal Requirements:

2. For studies involving third-party data, we encourage authors to share any data specific to their analyses that they can legally distribute. PLOS recognizes, however, that authors may be using third-party data they do not have the rights to share. When third-party data cannot be publicly shared, authors must provide all information necessary for interested researchers to apply to gain access to the data. (https://journals.plos.org/plosone/s/data-availability#loc-acceptable-data-access-restrictions)

a) A description of the data set and the third-party source

b) If applicable, verification of permission to use the data set

c) Confirmation of whether the authors received any special privileges in accessing the data that other researchers would not have

d) All necessary contact information others would need to apply to gain access to the data

3. We note that Figure S1 in your submission contain [map/satellite] images which may be copyrighted. All PLOS content is published under the Creative Commons Attribution License (CC BY 4.0), which means that the manuscript, images, and Supporting Information files will be freely available online, and any third party is permitted to access, download, copy, distribute, and use these materials in any way, even commercially, with proper attribution. For these reasons, we cannot publish previously copyrighted maps or satellite images created using proprietary data, such as Google software (Google Maps, Street View, and Earth). For more information, see our copyright guidelines: http://journals.plos.org/plosone/s/licenses-and-copyright.

a. You may seek permission from the original copyright holder of Figure S1 to publish the content specifically under the CC BY 4.0 license. 

Reviewers' comments:

Reviewer's Responses to Questions

**Comments to the Author**

1. Is the manuscript technically sound, and do the data support the conclusions?

Reviewer #1: No

Reviewer #2: Yes

2. Has the statistical analysis been performed appropriately and rigorously? 

Reviewer #1: No

Reviewer #2: Yes

3. Have the authors made all data underlying the findings in their manuscript fully available?

Reviewer #1: Yes

Reviewer #2: Yes

4. Is the manuscript presented in an intelligible fashion and written in standard English?

Reviewer #1: Yes

Reviewer #2: No

5. Review Comments to the Author

Reviewer #1: Review of PONE-D-23-24394: Environmental effects on reproductive performance in preindustrial women

I appreciated the opportunity to review the manuscript “Environmental effects on reproductive performance in preindustrial women.” The authors use an historic database of individuals and family histories from the Registre de la population du Québec Ancien (RPQA) to study the effects of moving between urban and rural environments in different regions of pre-industrial Montreal.

Given the authors are studying humans, not animal populations, with this analysis, it is not clear that the hypotheses and statistical analyses are appropriate. These limitations prevent drawing definitive conclusions from the analysis. Below I summarize my major comments on the manuscript.

Environmental conditions:

1. The authors use a very crude classification of environmental conditions: rural-south, rural-north, urban-south, and urban-north and make generalizations regarding how “good” or “bad” the conditions are. With such limited information on the actual environmental conditions, it is not clear how much we actually learn from these classifications and the normative statements are not well-defended with evidence.

2. The authors argue that their approach of using fixed locations (the 4 groupings mentioned above) offers advantages over a temporal approach that examines changing conditions over time. I would argue that an approach that allows for both spatial and temporal variation is preferred. The authors include women born between 1640-1729/1750, thus spanning over 100 years. To say that the temporal changes do not matter over such a long period is incorrect. Particularly when trying to say something regarding urban areas, which undoubtedly changed substantially during that period. While the authors include year of birth to control for cohort effects, this does not account for temporal changes that affect all cohorts in the same time period (but across ages). For example, a famine in a given year may affect the age at first birth for younger cohorts and also the lifetime number of children born for all cohorts.

3. It was unclear from the manuscript how the authors are tracking the age and period within their life course fertility at which people move locations, which undoubtedly matters. For example, do we know whether the “switchers” switched before their first birth? If not, then the authors are associating the wrong conditions with that outcome.

Individual-level determinants of fertility and human behavior:

1. The authors ascribe the environmental conditions of a given parish (rural-south, rural-north, urban-south, and urban-north) to all of the individuals that live in that parish. Even if certain areas tended to be better agriculturally than others, not all individuals fared equally well. It is most certainly better to be a wealthy individual in a “bad” environment than a poor person in a “good” environment. While the RPQA data does not include information on individual or household wealth, it would improve the hypotheses and statistical analysis for the authors to take seriously the role of individual-level variation.

2. Relatedly, while the authors are using a human population to make analogous comparisons to animal populations (e.g., red deer, elephants, etc), they are missing the seminal literature on determinants of fertility (and disentangling individual, social, behavioral, environmental factors), as well as some seminal literature on early life environments. Reviewing this literature would help to inform the hypotheses the authors test in this analysis and ensure that they are appropriate for studying humans rather than animals. For example, in social science and demography fields it is understood that in the early stages of the fertility transition (which typically corresponds to more rural, pre-industrial times).

Almond, Douglas, and Janet Currie. “Killing Me Softly: The Fetal Origins Hypothesis.” Journal of Economic Perspectives 25, no. 3 (August 1, 2011): 153–72. https://doi.org/10.1257/jep.25.3.153.

Becker, Gary S., and H. Gregg Lewis. “On the Interaction between the Quantity and Quality of Children.” Journal of Political Economy 81, no. 2, Part 2 (1973): S279–88. https://doi.org/10.1086/260166.

Bongaarts, John. “Does Malnutrition Affect Fecundity? A Summary of Evidence.” Science 208, no. 4444 (1980): 564–69.

Bongaarts, John. “Modeling the Fertility Impact of the Proximate Determinants: Time for a Tune-Up.” Demographic Research 33 (September 11, 2015): 535–60. https://doi.org/10.4054/DemRes.2015.33.19.

Bongaarts, John, and Dennis Hodgson. Fertility Transition in the Developing World. SpringerBriefs in Population Studies. Cham: Springer International Publishing, 2022. https://doi.org/10.1007/978-3-031-11840-1.

Friedlander, Dov. “Demographic Responses and Population Change.” Demography 6, no. 4 (November 1, 1969): 359–81. https://doi.org/10.2307/2060083.

Lesthaeghe, Ron. “The Second Demographic Transition: A Concise Overview of Its Development.” Proceedings of the National Academy of Sciences 111, no. 51 (December 23, 2014): 18112–15. https://doi.org/10.1073/pnas.1420441111.

3. The authors make several normative statements regarding a younger age at first birth and higher number of offspring being good (e.g., 175-176). While these may represent better reproductive performance in animal populations, it is well established in the human health literature that early age at first birth (often defined as 15 years old or younger) is associated with bad maternal and child health outcomes, such as low birthweight, preterm birth, infant mortality, and maternal mortality. In demographic literature, which emphasizes concepts of reproductive autonomy, it is not the total number of children born to a woman that matters but rather whether she is able to attain her desired family size. In many cases, her desired family size is lower than the number of children she has (so more is not unequivocally better), due to complex and intersecting social, cultural, and economic reasons. Beyond the issue with calling younger age at first birth and higher number of offspring “good”, this framing ignores (in the statistical sense) how these outcomes are in fact correlated, particularly for human populations, which are presumably quite different from red deer, elephants, and other animal populations in ways not acknowledged by the authors. As a result, certain findings are expressed as surprising, when in fact they are consistent with expectation for humans. For example, the authors state (374-379) that given the correlation between younger AFR and higher NO it was surprising that women born in urban (“bad”) parishes didn’t have more children surviving to adulthood (which they define as 15 years of age). Given younger AFR is associated with lower chances of child survival and a well-established reason for large numbers of pregnancies and children in preindustrial, pre-demographic transition times is to protect against child mortality and have a sufficiently large family to work in farms and other household activities.

Statistical Analyses:

1. While the procedure to remove insignificant terms from statistical models is appropriate in some fields and for some analyses, in this case as the authors wish to test competing hypotheses, any deletion of terms prevents their ability to do so in a systematic and comprehensive manner. If the authors have a theory about which variables determine these fertility outcomes, those variables should all be in the model regardless of statistical significance because any lack of statistical significance informs the interpretation and conclusion regarding the hypotheses.

2. To truly test these different hypotheses, I would suggest introducing the early life, later life, and switching environmental variables in sequentially and then using all combinations. This would allow the authors, for example, to see what happens to early life environmental variables once they control for switching or once they control for later life. Because this approach would also involve a large number of statistical tests, it would be important for the authors to account for multiple hypotheses testing to ensure they are not drawing conclusions from type I errors.

3. A more minor comment on the interpretation of results is that the authors should be careful in how they interpret terms on categorical variables and make sure they are always interpreted as relative to the dropped category. For example, as stated in the notes for Table 2 the reference level is rural south for birth environment, which means the coefficients on rural north, urban south, and urban north are all interpreted as relative to rural south. But in 216-218, the authors interpret the coefficients on the urban variables and relative to both rural variables.

Reviewer #2: Dear Editor: I appreciate the opportunity to review the article entitled”Environmental effects on reproductive performance in preindustrial women"

I think there are some flaws in each portion of the work. Here are my professional observations of the manuscript and any suggestions I might have:

The introduction section was well-written and described. Further, the author’s need to incorporate the severity and consequence of the problem

The authors fail to clearly operationalize variables;

What are the strengths and limitations of your study?

The discussion section was well-written and should be elaborated more than the current description in regarding with adding valuable knowledge to global literature and future investigations.

The conclusion section should be rewritten as per the result of the study and feasible for the implementation.

6. PLOS authors have the option to publish the peer review history of their article (what does this mean?). If published, this will include your full peer review and any attached files.

Reviewer #1: No

Reviewer #2: No

---

## [Author Response · Author response to Decision Letter 0]

30 Jul 2024

Dear Reviewers,

Thank you very much for your thoughtful and constructive feedback on our manuscript entitled, "Early and adult life environmental effects on reproductive performance in preindustrial women." We are grateful for your detailed comments, which have been invaluable in improving our work.

We especially appreciated your suggestions on better justifying how demographic data on humans could complement wildlife eco-evolutionary questions. This insight has significantly enhanced the manuscript's relevance and depth. Additionally, we have thoroughly revised the manuscript to address all your comments and ensure consistency and flow throughout the document. While the major changes are highlighted in the track change version, we have also made several minor adjustments for clarity and coherence.

In the document "Response to Reviewers," we provide a detailed, point-by-point response to each of your comments, indicated in italics. Each response is numbered consecutively to facilitate cross-referencing and review.

We thank you once again for your valuable feedback and consideration of our revised submission. We look forward to hearing from you in due course.

Yours sincerely,

Lidia Colejo Durán, on behalf of all co-authors

---

## [Editor Report · Decision Letter 1]

16 Aug 2024

Early and adult life environmental effects on reproductive performance in preindustrial women

PONE-D-23-24394R1

Dear Madam,

We’re pleased to inform you that your manuscript has been judged scientifically suitable for publication and will be formally accepted for publication once it meets all outstanding technical requirements.

Kind regards,

Edison Arwanire Mworozi, M.D

Academic Editor

PLOS ONE

I have no further contributions to make towards this paper. 

Thank you and good luck in your endeavors.

To the PLOS Board, please do not send this paper to me again. Thank you.

---

## [Editor Report · Acceptance letter]

27 Aug 2024

PONE-D-23-24394R1 

PLOS ONE

Dear Dr. Colejo-Durán, 

I'm pleased to inform you that your manuscript has been deemed suitable for publication in PLOS ONE. Congratulations! Your manuscript is now being handed over to our production team.

Kind regards, 

on behalf of

Professor Edison Arwanire Mworozi 

Academic Editor

PLOS ONE